# Factors associated with shorter length of admission among people with dementia in England and Wales: retrospective cohort study

Rahil Sanatinia ![ORCID],[1] Alistair Burns,[2] Peter Crome,[3] Fabiana Gordon,[4] Chloe Hood,[5] William Lee,[6] Alan Quirk,[7] Kate Seers,[8] Sophie Staniszewska,[8] Gemma Zafarani,[1] Mike Crawford[1]

For numbered affiliations see end of article.

**Correspondence to**
Dr Rahil Sanatinia;
r.sanatinia@imperial.ac.uk

## ABSTRACT

**Objectives** To identify aspects of the organisation and delivery of acute inpatient services for people with dementia that are associated with shorter length of hospital stay.

**Design and setting** Retrospective cohort study of patients admitted to 200 general hospitals in England and Wales.

**Participants** 10 106 people with dementia who took part in the third round of National Audit of Dementia.

**Main outcome measure** Length of admission to hospital.

**Results** The median length of stay was 12 days (IQR=6–23 days). People with dementia spent less time in hospital when discharge planning was initiated within 24 hours of admission (estimated effect −0.24, 95% CI: −0.29 to −0.18, p<0.001). People from ethnic minorities had shorter length of stay (difference −0.066, 95% CI: −0.13 to −0.002, p=0.043). Patients with documented evidence of discussions having taken place between their carers and medical staff spent longer in hospital (difference 0.26, 95% CI: 0.21 to 0.32, p<0.001). These associations held true in a subsample of 669 patients admitted with hip fracture and data from 74 hospitals with above average carer-rated quality of care.

**Conclusions** The way that services for inpatients with dementia are delivered can influence how long they spend in hospital. Initiating discharge planning within the first 24 hours of admission may help reduce the amount of time that people with dementia spend in hospital.

## INTRODUCTION

The number of people with dementia who are admitted to acute hospitals is increasing.[1 2] The unfamiliar environment and disruption to daily routines that people with dementia experience when they are admitted to hospital can cause emotional distress and precipitate behavioural symptoms.[3] When people with dementia are admitted to hospital they need more care and are more likely to suffer functional decline.[4] Patients with dementia are more impaired and vulnerable than patients without dementia and are at greater risk of

## STRENGTHS AND LIMITATIONS OF THIS STUDY

⇒ We used data from 98% of all acute hospitals in England and Wales, so our results are generalisable across the country.
⇒ People with lived experience of dementia and caring for people with dementia helped us select items for including in the analysis and interpret the findings.
⇒ All data were observational, so we cannot know whether the associations we found are causal.
⇒ This research relied on audit data of variable quality.
⇒ We did not have data on liaison psychiatric services in a sixth of hospitals, limiting our ability to explore whether the provision of these services influences the length of stay of people with dementia.

adverse outcomes.[5] Compared with other inpatients, patients with dementia spend more time in hospital, are more likely to be readmitted within 28 days of their discharge,[6] and have significantly higher costs.[7 8]

A number of interventions have been developed to try to improve the quality of acute inpatient care for people with dementia. These efforts include the deployment of specialist nurses, the expansion of mental health liaison teams and the development of specialist inpatient units. A number of hospitals employ specialist dementia nurses to support the management of patients with dementia,[9] but it is unclear whether this influences the amount of time that people with dementia spend in hospital.[10] Specialist units aim to better meet the needs of inpatients with dementia[11] by changing the environment of the ward, training staff to deliver patient centred care and delivering a programme of organised therapeutic and diversionary activities.[12] While carers of patients admitted to these units report improved satisfaction, there is no evidence to

date that they impact on the amount of time that people stay in hospital.[13]

Tadros *et al* reported that the implementation of a new integrated psychiatric liaison service reduced the amount of time that elderly patients admitted to City Hospital, Birmingham (UK), spent in hospital.[14] These findings prompted an expansion of mental health liaison services nationwide,[15] but the findings of this study are yet to be replicated. Carer involvement has been identified as crucial to better patient experience during hospital admission. To achieve this, family carers need to be kept informed and provided with support and help when necessary.[16]

National audits examining acute care for people with dementia have highlighted considerable variation in the quality of care that hospitals deliver to people with dementia,[17 18] but little is known about reasons for these differences. Therefore, our objective was to identify aspects of the organisation and delivery of care for inpatients with dementia that are associated with shorter length of stay. This analysis was conducted as part of a wider study that examined the quality of care for inpatients with dementia.[19] As part of this mixed-methods study, we interviewed ward staff, managers and carers of people with dementia to explore factors that influence the amount of time that people stay in hospital.

## METHODS
### Study setting and sample
We analysed data from a retrospective cohort of patients who were enrolled in the third round of National Audit of Dementia.[17] The audit aimed to evaluate the quality of care provided by all acute hospitals in England and Wales. We used data from three components of the audit: a hospital-level organisational checklist; a retrospective case note audit with a target of a minimum of 50 sets of patient notes of patients who had been given a clinical diagnosis of dementia and been admitted to hospital for 72 hours or longer between April 2016 and November 2016; and a survey of carer experience of quality of care. We combined these with the data from the second round of national survey of Liaison Psychiatry services in England, which collected data on mental health liaison service provision in all the acute hospitals in England.[20]

### Study measures
We calculated the length of admission for each audit participant using their date of admission and date of discharge. To select independent variables for the analysis from the wide range included in the audit, we identified factors which may influence length of stay based on a scoping review of the literature.[19] We presented this list to the members of the Project Steering Group for their feedback, including people with lived experience of dementia and carers of people with dementia. In the final stage, we excluded any variables present in over 90% of hospitals. The lists of included variables and those excluded,

and source of data, are presented in online supplemental appendices 1 and 2.

### Data management and analysis
Some of the predictor variables had multiple categories with small numbers in each category, we therefore combined these to obtain stable results[21] (details are available on request). Primary diagnosis/cause of admission was taken as the first reason entered on the case note audit. Case notes included over 100 categories of primary diagnosis. We grouped these together in 11 categories such that myocardial infarction was combined with other vascular conditions and kidney and urological conditions were grouped together.

We conducted univariate analyses to explore the association between the dependent variable (length of stay) and patient-level (age, gender, primary diagnosis), ward-level (type of ward) and hospital-level (access to liaison mental health services, deployment of specialist dementia nurses, involvement of the trust board) predictor variables. We used multiple linear regression for our analysis. Given the nested structure of the data, the final analysis was carried out using hierarchical models. The main analysis was conducted on the complete data, so missing data were not given any treatment. All data were analysed using statistical packages STATA (V.13) and SPSS V.23.

The outcome measure, length of admission, was skewed so the logarithm of length of stay was used. We excluded all those who died during their admission from all analyses.

In order to test the robustness of associations found in the main analysis, we conducted two sensitivity analyses. In the first sensitivity analysis we restricted the samples to patients who were admitted with hip fractures/trauma. In the second sensitivity analysis, we restricted the sample to patients who were admitted to hospitals that scored above the median level of carer satisfaction in the carer survey that was conducted as part of the audit.[17] This was to address concerns raised by members of our Steering Group that hospitals could achieve shorter length of stay by prematurely discharging patients with dementia. We therefore conducted this sensitivity analysis to explore whether factors associated with shorter length of stay were not at the cost of lower levels of care satisfaction. Researchers who conducted the analysis of data were not involved in collecting and cleaning of data.

### Patient and public involvement
We worked closely with the Steering Group for the National Audit of Dementia, which included people with lived experience of dementia and caring for people with dementia. In addition, the Project Management Group for the study included a member with dementia and a carer representative. Patients and carers contributed to the study by helping us select items for including in the analysis from the long list that we had originally developed. They also contributed to interpretation of study findings. Some of the patients involved in the early stages

of the study were not able to contribute to the later stages due to the progression of their condition. One of the carer representatives (GZ) helped us prepare this manuscript and write the lay summary for the study.

## RESULTS

In total, 200 (98.5%) of 203 acute hospitals in England and Wales took part in the audit. Organisational checklists were returned from all 200 hospitals along with data on 10 106 patient records and 4688 carer questionnaires. The mean age of participants was 84.3 (SD=7.9), 8274 (81.9%) were white participant, and 6054 (59.9%) were women. Descriptive data on the predictor variables from the audit are presented in online supplemental online supplemental appendices 3, 4 and 5.

### Liaison psychiatry data

Data for the 2015 audit of psychiatric liaison services were returned by teams serving 176 (88%) of the 200 hospitals

that took part in the national audit of dementia. We used data on the two predictor variables in our analysis; the number of hours covered by the liaison service, and whether the team included an old age psychiatrist. Ninety-one (53.53%) liaison services had 24-hour coverage, 72 (42.35%) were available for more than 40 hours per week and 4 (2.35%) for less than 40 hours per week. In total, 83 of 169 (49.11%) liaison services included an old age psychiatrist. Full details can be found in online supplemental appendix 6.

### Length of stay

Valid data on the length of stay were available for 10 105 patients with dementia; the median length of stay was 12 days (IQR=6–23 days). There was variation in the median length of stay across the 200 hospitals, ranging from 5 to 39 days (IQR=10–14 days).

The results of the multivariate analyses are presented in table 1. The following variables were associated with

**Table 1** Multivariate analysis of factors associated with length of stay among 8817 patients with dementia treated at 196 acute hospitals

| Predictor variable | Interactions* | Estimated effect (95% CI) | P value |
|---|---|---|---|
| Patient age | | −0.003 (−0.005 to 2.19) | 0.052 |
| Patient ethnicity—white | | − | |
| Black, Asian and minority ethnic | | −0.066 (−0.13 to −0.002) | 0.043 |
| Ward type—care of the elderly | | − | |
| Cardiac | | −0.33 (−0.47 to −0.19) | <0.001 |
| General medical | | −0.28 (−0.33 to −0.22) | <0.001 |
| Orthopaedics | | −0.16 (−0.26 to −0.05) | 0.003 |
| Surgical | | −0.32 (−0.41 to −0.23) | <0.001 |
| Other medical | | −0.26 (−0.33 to −0.18) | <0.001 |
| Other | | −0.56 (−0.74 to −0.37) | <0.001 |
| Primary diagnosis—respiratory | | − | |
| Fall | | 0.24 (0.17 to 0.32) | <0.001 |
| Orthopaedic | | 0.39 (0.27 to 0.50) | <0.001 |
| Delirium/confusion | | 0.25 (0.18 to 0.33) | <0.001 |
| Other | | 0.13 (0.05 to 0.20) | 0.001 |
| Discharge planning within 24 hours of admission Yes versus no | | −0.24 (−0.29 to −0.18) | <0.001 |
| Executive board reviews delayed discharge Yes versus no | | '−0.08 (−0.17 to 0.006) | 0.069** |
| Evidence of discussing discharge with carers Yes versus no | | 0.26 (0.21 to 0.32) | <0.001 |
| Evidence of discussing discharge with consultant | In hospitals with old age liaison psychiatrist Yes versus no | | |
| Yes | | −0.039 (−0.12 to 0.05) | 0.367 |
| No | | −0.15 (−0.03 to −0.26) | 0.012 |

*When an interaction effect is present, the estimated effects of risk factors are interpreted taking into effect the interaction column.

shorter average length of hospital stay: presenting with a respiratory condition, black, Asian and minority ethnic (BAME) background, initiating the discharge planning within 24 hours of admission, and patients for whom there was not a record of discussion with the responsible consultant (only in hospitals with a specialist old age psychiatrist). In hospitals where trust boards regularly review delayed discharges, there was a non-statistically significant trend towards patients having shorter hospital stays.

Variables associated with longer average length of stay were; type of ward that patients were being treated on with those on the care of the elderly having longer average hospital stays compared with general medical and surgical wards, and discussions with carers being recorded in patients' case notes.

All risk factors were included in the initial full-effects model. This model was then reduced by excluding stepwise non-significant effects. The model presented here is the model with only the significant effects.

### Sensitivity analyses

Results from two sensitivity analyses are presented in online supplemental appendices 7 and 8. We looked at factors associated with length of stay among 669 patients treated for hip fracture at 170 acute hospitals. We also performed multivariate analysis of factors associated with length of stay among 3375 patients treated at 74 acute hospitals with higher carer-rated satisfaction. There were fewer associations in the first sensitivity analysis when we restricted the sample to patients who were admitted to hospital due to hip fracture or related injuries (n=669). The variables associated with shorter average length of stay were: documented evidence of initiating discharge planning within 24 hours of admission, BAME background (only among women but not men), and female patients in those hospitals that deployed dementia nurse specialists. When there was documented evidence that discharge had been discussed with carers, patients had longer hospital stays.

In the analysis of data from the subsample of patients admitted to the 74 hospitals that had carer-rated satisfaction scores above the median (n=3375), primary diagnosis and type of ward were still associated with length of stay. Patients with respiratory conditions had shorter hospital admissions compared with other diagnoses, including cardiac/vascular, fall and hip fracture. Those on the care of the elderly wards seemed to have longer lengths of stay compared with patients being treated on general medical, cardiac, orthopaedics and surgical wards. When there was no documented evidence of discussing discharge with carers, female patients had shorter hospital admissions. In hospitals where the liaison service included an old age specialist consultant psychiatrist, patients had shorter hospital stays when there was evidence that discharge planning had been initiated within the first 24 hours of admission, as well as in hospitals where there were not any social workers or designated persons. In situations when there was evidence of discussing discharge with the consultant responsible for patient's care, average length of stay was higher. Table 2 presents a summary of results from the main multivariate and two sensitivity analyses.

### DISCUSSION

We aimed to identify measures that are associated with shorter hospital admissions for patients with dementia. In the multivariate analysis, in addition to primary diagnosis and the type of ward that patients were admitted to, we found that the ethnicity, discharge planning within 24 hours of admission and documented discussions about discharge with carers were associated with the amount of time that patients spent in hospital. While presenting with a respiratory condition, BAME background, and early discharge planning were associated with shorter hospital stays, documented discussions with carers was associated with longer average hospital stays. These associations were supported by sensitivity analysis of patients admitted with a hip fracture and in hospitals with higher carer-rated quality of care for people with dementia.

Of the modifiable variables, planning discharge within 24 hours of admission had the strongest negative association with length of hospital stay (Coefficient=−0.24).

These data support those of a smaller scale study of 85 patients who spent more than 14 days in an acute hospital in Ireland, which identified timely discharge planning and clear communication as measures for preventing prolonged hospital stays.[22] Support for early discharge planning also comes from the results of a series of case studies conducted by National Health Service trusts across England as part of their quality improvement programmes (https://improvement.nhs.uk/resources/guide-reducing-long-hospital-stays/).

Patients with dementia are at increased risk of mortality and morbidity when hospitalised, and health professionals are usually faced with high levels of complex care needs when managing care and discharge of this patient group.[23] In a literature review of barriers and facilitators to discharge planning for people with dementia, involving family carers in the early stages of admission was identified to improve the discharge process.[24]

In the wider mixed methods study, we carried out interviews with clinicians and senior managers to shed light on the associations found in quantitative data analysis. In these interviews, we found discharge planning among patients with dementia is not straightforward. Staff told us that despite the complexity, setting a planned discharge date as soon as preliminary investigations are carried out and working out the potential barriers to a timely discharge are important.[19] Senior nurses told us that allocating tasks to specific members of staff and daily review of the progress would ensure that problems are identified early and escalated to relevant departments.

In our interviews, we also explored the association found between carer involvement and longer admissions. There was a consensus that involving family carers helps

**Table 2** Summary of results from the main multivariate and two sensitivity analyses of modifiable predictor variables

| Predictor variable | Interactions* | Main multivariate analysis | Admitted with hip fracture (n=669) | Admitted to hospitals with higher carer satisfaction (n=3375) |
|---|---|---|---|---|
| | | Estimated effect (95% CI) P value | | |
| Discharge planning within 24 hours of admission Yes versus no | | −0.24 (−0.29 to −0.18) <0.001 | −0.38 (−0.55 to −0.21) <0.001 | |
| Yes | Old age liaison psychiatrist consultant Yes versus no | | | −0.29 (−0.46 to −0.11) 0.002 |
| No | | | | −0.07 (−0.25 to 0.11) 0.425 |
| Evidence of discussing discharge with carers Yes versus no | | 0.26 (0.21 to 0.32) <0.001 | 0.41 (0.27 to 0.55) <0.001 | |
| Yes | Gender of the patient Female versus male | | | 0.005 (−0.07 to 0.08) 0.901 |
| No | | | | −0.20 (−0.34 to −0.05) 0.008 |

*When an interaction effect is present, the estimated effects of risk factors are interpreted taking into effect the interaction column.

patients settle and adhere with treatments when admitted to hospital. Family carers usually acknowledge that long hospital stays are not conducive to patients' well-being. However, lack of clear communication between family carers and hospital staff from the outset may mean that expectations are not shared, which might lead to longer admissions to hospital. In a previous literature review of barriers and facilitators to discharge planning for people with dementia, researchers identified involving family carers in the early stages of admission to improve discharge process.[24]

In our univariate analysis, we found that people admitted to specialist care of the elderly wards spent longer in hospital, than those admitted to general medical and surgical wards. However, ward type did not predict length of stay once the effect of demographic and clinical factors was taken into account in the multivariate analysis. These findings may reflect the complex mental, physical and social needs of people admitted to specialist wards.[25]

The association between BAME background and shorter length of stay was unexpected and require further investigation.

### Strengths and weaknesses of the study

We used data from the third round of the National Audit of Dementia from 98% of all acute hospitals in England and Wales. This means that our results could be generalisable across the country. Data were complete on length of stay, and level of data completeness was high for key variables.

In this study, we had the advantage of involvement of people with lived experience of dementia and carers of people with dementia in various stages of study planning, data analysis and interpretation. The study has several limitations. We relied on the information submitted to the audit team by hospital staff on whether hospitals had adopted a strategy or not. We cannot rule out the possibility that in some instances, certain assessments or discussions might have taken place but not been recorded. While the large dataset makes the findings more generalisable, we cannot be certain if the associations we found are causal. For instance, our finding that patients whose records included documented evidence of discussions with carers having longer admissions could be causal, that is, that discussions with carers raised additional issues that they felt needed to be addressed before the person they cared for was discharged home. However, it could also be confounded by complexity and coexisting conditions of patients that is, staff needed to liaise more with carers of patients with complex conditions, and such patients also required a longer period of treatment in hospital. It is possible that documenting discussions with carers happens more often in complex cases, where there has been more deterioration, or there is need for change of residence.[26]

There can be a trade-off between shorter hospital stay and other aspects of quality of care. Our finding that the association between shorter length of stay and early discharge planning was also present in hospitals with higher carer-rated quality of care was reassuring.

Our findings reflect practice from 2016. Since these data were collected, the COVID-19 pandemic has further highlighted the importance of discharge planning to ensure that patient safety is maintained, and premature discharge avoided. Discharge planning at the time of COVID-19 pandemic has been even more complicated as care homes might be unwilling to have patients back immediately after discharge from acute hospitals. Concerns about COVID-19 outbreaks among care home residents mean that family carers are reluctant for their loved ones to return to care homes.[27 28]

Previous research has raised the possibility that psychiatric liaison services might enable more timely discharge of older adults with mental health conditions including dementia.[14 29] Missing data on availability of psychiatric liaison services meant that we could not fully examine the potential impact of this factor on length of stay.

### Implications for practice and future research

We recommend that hospitals put systems in place for planning discharge as soon as possible, regardless of the complexity of patient presentation. Clear communication with family carers from early stages of admission helps them form realistic expectations to avoid unnecessary delays in discharge.

Future research should explore the unexpected finding of shorter hospital stays among patients with dementia of from BAME backgrounds.

**Author affiliations**
¹Division of Psychiatry, Imperial College London, London, UK
²Division of Neuroscience and Experimental Psychology, The University of Manchester, Manchester, UK
³Research Department of Primary Care and Population Health, University College London, London, UK
⁴Population Health Sciences, University of Bristol, Bristol, UK
⁵College Centre for Quality Improvement, Royal College of Psychiatrists, London, UK
⁶College of Medicine and Health, University of Exeter, Exeter, UK
⁷CCQI, Royal College of Psychiatrists, London, UK
⁸Warwick Research in Nursing, University of Warwick Warwick Medical School, Warwick, UK

**Acknowledgements** We acknowledge the valuable contribution made by our collaborators: Daphne Wallace (retired psychiatrists with lived experience), Amy Claringbold (trial manager, Imperial College London) and Sara Hammond who led the scoping literature review and helped with the interpretation of the results. We would also like to thank patients and carers and other members of the Steering Group, including: Nicci Gerrard (carer representative, John's Campaign), Sue Pierlejewski (carer representative), Hilary Doxford (Alzheimer's Society Ambassador), Jayne Goodrick (carer representative) and Chris Roberts (Alzheimer's Society Ambassador).

**Contributors** RS was the lead researcher, assisted with the analysis of quantitative data and prepared the final manuscript. AB advised on the methods. PC chaired the Steering Group and contributed to the interpretation of the results. FG led the analysis of quantitative data. CH advised on the methods and contributed to the interpretation of the results. WL provided advice and support on the analysis of data

from the second national survey of Mental Health Liaison Services. AQ contributed to the interpretation of the results. KS contributed to the interpretation of the results. SS contributed to the interpretation of the results. GZ contributed to the interpretation of the results. MC was the chief investigator and oversaw all aspects of the study. All authors read, made comments and approved the final manuscript. The guarantor is RS.

**Funding** The work was supported by the Health Services and Delivery Research programme of the National Institute for Health Research (NIHR) (HS&DR 14/154/09). Infrastructure support for this research was provide by the NIHR Imperial Biomedical Research Centre.

**Competing interests** Institutions for all authors have received funding from National Institute for Health Research (NIHR) for other studies. AQ and CH work at the Royal College of Psychiatrists and oversee the National Audit of Dementia. MC is director of the College Centre for Quality Improvement at the Royal College of Psychiatrists and has been a member of the NIHR HTA General Committee. PC was chair of the Steering Group for the National Audit of Dementia and has been on the HTA PCCPI Panel and HTA Prioritisation Committee A (Out of Hospital). KS has been a member of the HS&DR Commissioning Board (researcher led). SS has been on the following committees: HS&DR Associate Board Members; HS&DR Researcher Led—Associate Board Members; INVOLVE BOARD. All authors declare no other competing interests.

**Patient consent for publication** Not applicable.

**Ethics approval** We received approval for the secondary analysis of data from the National Clinical Audit and Patient Outcomes Programme from the Healthcare Quality Improvement Partnership prior to the start of study (Reference: HQIP162).

**Provenance and peer review** Not commissioned; externally peer reviewed.

**Data availability statement** Data are available upon reasonable request. Data may be obtained from a third party and are not publicly available. All available data can be obtained by contacting the corresponding author. All data requests should be submitted to the corresponding author for consideration. Access to anonymised data may be granted following review. Applications for access to data from the third round of the National Audit of Dementia should be made to the Healthcare Quality Improvement Partnership. Details of the process for obtaining these data are available at: https://www.hqip.org.uk/national-programmes/accessing-ncapop-data/#.W-7gS0ca7oo .

**ORCID iD**
Rahil Sanatinia http://orcid.org/0000-0002-3466-4644

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
