## [Reviewer comments · BMJ Open]

ARTICLE DETAILS

TITLE (PROVISIONAL)	Factors associated with shorter length of admission among people with dementia in England and Wales: Retrospective cohort study.
AUTHORS	Sanatinia, Rahil; Burns, Alistair; Crome, Peter; Gordon, Fabiana; Hood, Chloe; Lee, William; Quirk, Alan; Seers, Kate; Staniszewska, Sophie; Zafarani, Gemma; Crawford, Mike

VERSION 1 – REVIEW

REVIEWER	Fox, Amanda Queensland University of Technology
REVIEW RETURNED	12-Jan-2021

GENERAL COMMENTS	Thank you for submitting this report of a sub-study (secondary analysis of data from a larger parent study) for review. I would recommend that you consider the following points for revision of your manuscript. 1. Abstract- p4 line 28 states that the patient has discussed with carers. Presume you mean carers of patients diagnosed with dementia who has discussed discharge with consultants or clinicians.2. Introduction the introduction is brief and relies on some dated literature to report broad concepts. It does not provide any background to literature specific to models of care or organisational aspects such as the ones you have explored in your study. Consider more recent literature on acute care for patients with dementia and models of care to assist in justifying this study.3. Methods I believe in its current presentation, it would be difficult to replicate this study due to lack of clarity of process and terms, variables used. -It is not clear to me if the data from the Psychiatric services survey included all the same hospitals as the retrospective audit. What variables were collected from each of these sources? - p7 line 16- you report that variables were excluded as they were not used by 90% of hospitals however, I cannot see any of these on the supplementary table -justify why you used the patients who scored above median level of carer satisfaction. I would have thought that those who scored below may have provided more useful information - variables collected need to be defined and how these were determined reported. eg. was the primary diagnosis based on coding? or written in patients notes by the consultant. p7 line 26 claims presenting complaint whilst in other areas of your paper it
---

refers to primary diagnosis. What constituted discharge planning commenced? Discharge discussed with carer- was that only with the primary care giver responsible for care of the patient with dementia or any carer and was it only with medical staff or any staff? how was this identified ie written in chart or in a check list. Where groups have been categorised, how and who determined this should be provided for replication. eg. ward type, primary diagnosis.

- p8 line 5 repetitive

4. Results

Again difficult to identify if those included from both sources were the same hospitals- liaison service collected in 2015, when was the audit conducted? how you are integrating this data to present findings is not clear.

Your results should be written to ensure that the reader is aware you are talking specifically about patients with a diagnosis of dementia.

5. Discussion

This section requires significant work and is very confusing. It appears to report findings from the larger study and has minimal if any comments related to this study. Discussion claims to have reviewed policies and practices - there has not been any review of policy in this research. Further, extensive discussion about interviews is undertaken however, this I presume occurred in the parent study.

The minimal background provided and inadequate review of the literature to justify this study is replicated and hindered the ability to write a meaningful discussion about this sub-study.

As with all large data set research the information available is generalisable but cannot provide details or reasons for any outcomes. This has been acknowledged in the limitations section, however, I also believe that overestimation of the significance of the outcomes in studies such as these can lead to incorrect assumptions. You may like to consider literature that looks at models of care to give insight into the reason that LOS was longer in the specific 'Care of the Elderly' wards. To do this you would need to explain what these wards are (methods section).

6. General comments

Throughout the manuscript there is a lack of correct use of capitals and mixed use of terminology that confuses the reader eg. use of Steering Group and Stakeholder Reference Group- are these the same? Consultant and Old Age Psychiatrist. This needs to be corrected throughout.

Tables need to be tidied to ensure consistent headings and that where * have been used - a descriptor is provided. Remove or amend empty sections of table.

I wish you all the best with your revision

REVIEWER

Mate, Karen
University of Newcastle, School of Biomedical Sciences and
Pharmacy

REVIEW RETURNED

01-Feb-2021

GENERAL COMMENTS

This retrospective cohort study examines the association of a number of patient-level, ward-level and hospital-level variables on the length of hospital stay for people with dementia. It addresses an important aspect of improving the care of people with dementia. The study design is sound, with input from key stakeholders including people with dementia and their carers. The manuscript is well written and clearly addresses the objectives, findings and limitations of the study.

A few issues that require attention prior to publication:

Methods

- Pg5 ln26 There is no mention of whether compliance with audit standards (discharge planning, discussing discharge with carer or consultant) is patient-level, ward-level or hospital level. From the data presented in Appendix 5, it appears to be collected at patient level but this should be specified in the methods. Also note that there is a discrepancy in the N values reported in Appendix 5 table title (N=10106) and the values in the table (all N=7385). These should be made consistent to either include or exclude the missing data.

- There is no specific explanation of how missing data were addressed. The predictor variables related to patients receiving care according to audit standards in particular, have high numbers of missing data. For example, "discharge planning within 24 hours of admission" has data missing for 26.9% of patients and N/A for 20.7% of patients. The authors should provide an explanation of the "N/A" category, whether these data were included in the analyses or not, and how missing data was managed in general.

Results

- Table 1: for interaction term (old age liaison psychiatrist), specify categories "yes vs no" or "present vs absent"

- Table 2: title should specify that the summary provided is for "organisation/delivery" or "modifiable?" predictors only; patient level and ward level factors are not included

Discussion

- The statement "Of these variables, planning discharge within 24 hours of admission had the strongest association with length of hospital stay." is not true. There are other non-modifiable or patient factors that have similar or stronger negative association with length of hospital stay (ie associated with shorter stay), and evidence of discussing discharge has similar positive association (ie associated with shorter stay). The statement is making a key point but needs to be modified so that it is correct eg "Of the modifiable (or organisational/delivery related) variables, planning discharge had the strongest negative association with length of hospital stay".

- Make the direction of associations more clear in the Discussion.

- Some of the Discussion of published literature (the authors own studies) is written as if it is part of the current study eg "in our interviews....." and "staff told us....." which is not appropriate.

This is particularly the case when dealing with the association between carer involvement and longer admissions. The authors should make a clear statement about the findings of the present study and then proceed to elaborate on the published literature (including their own) to provide possible explanations of the findings.

- It would be helpful for the authors to comment on the magnitude of the associations and whether they are of clinical significance in terms of the length of hospital stay.

Finally, there are few typos throughout the text eg
Pg 4 "symotms"
Pg10 "mor"
Pg11 "reqite"

VERSION 1 – AUTHOR RESPONSE

Reviewer: 1

Dr. Amanda Fox, Queensland University of Technology

1. Abstract- p4 line 28 states that the patient has discussed with carers. Presume you mean carers of patients diagnosed with dementia who has discussed discharge with consultants or clinicians.

Thank you- revised

2. Introduction

the introduction is brief and relies on some dated literature to report broad concepts. It does not provide any background to literature specific to models of care or organisational aspects such as the ones you have explored in your study. Consider more recent literature on acute care for patients with dementia and models of care to assist in justifying this study.

Thanks, amended and added more specific literature

3. Methods

I believe in its current presentation, it would be difficult to replicate this study due to lack of clarity of process and terms, variables used.

-It is not clear to me if the data from the Psychiatric services survey included all the same hospitals as the retrospective audit. What variables were collected from each of these sources? **The details are presented in results section:**

Data for the 2015 audit of psychiatric liaison services were returned by teams serving 176 (88%) of the 200 hospitals that took part in the national audit of dementia. We used data on the two predictor variables in our analysis; the number of hours covered by the liaison service, and whether the team included an old age psychiatrist. Full details can be found in online supplemental appendix 6.

Please advise if you prefer to move it to the methods section

- p7 line 16- you report that variables were excluded as they were not used by 90% of hospitals however, I cannot see any of these on the supplementary table

Thank you, amended the sentence and details are in appendix 2 in the supplementary file

-justify why you used the patients who scored above median level of carer satisfaction. I would have thought that those who scored below may have provided more useful information

Thanks, noted and elaborated further on the rationale in the text

- variables collected need to be defined and how these were determined reported. eg. was the primary diagnosis based on coding? or written in patients notes by the consultant. p7 line 26 claims presenting complaint whilst in other areas of your paper it refers to primary diagnosis. What constituted discharge planning commenced? Discharge discussed with carer- was that only with the primary care giver responsible for care of the patient with dementia or any carer and was it only with medical staff or any staff? how was this identified ie written in chart or in a check list. Where groups have been categorised, how and who determined this should be provided for replication. eg. ward type, primary diagnosis.

Thanks, noted and added more details to the text and on the supplementary document

- p8 line 5 repetitive

4. Results

Again difficult to identify if those included from both sources were the same hospitals- liaison service collected in 2015, when was the audit conducted? how you are integrating this data to present findings is not clear.

Your results should be written to ensure that the reader is aware you are talking specifically about patients with a diagnosis of dementia.

We have already provided the details in the methods:

'We used data from three components of the audit: a patient who had been given a clinical diagnosis of dementia and been admitted to hospital for 72 hours or longer between April 2016 to November 2016.'

And about liaison service below in the results highlighted in yellow in the marked copy

Data for the 2015 audit of psychiatric liaison services were returned by teams serving 176 (88%) of the 200 hospitals that took part in the national audit of dementia. We used data on the two predictor variables in our analysis; the number of hours covered by the liaison service, and whether the team included an old age psychiatrist. Full details can be found in online supplemental appendix 6.

5. Discussion

This section requires significant work and is very confusing. It appears to report findings from the larger study and has minimal if any comments related to this study. Discussion claims to have reviewed policies and practices - there has not been any review of policy in this research. Further, extensive discussion about interviews is undertaken however, this I presume occurred in the parent study.

The minimal background provided and inadequate review of the literature to justify this study is replicated and hindered the ability to write a meaningful discussion about this sub-study.

As with all large data set research the information available is generalisable but cannot provide details or reasons for any outcomes. This has been acknowledged in the limitations section, however, I also believe that overestimation of the significance of the outcomes in studies such as these can lead to incorrect assumptions. You may like to consider literature that looks at models of care to give insight into the reason that LOS was longer in the specific 'Care of the Elderly' wards. To do this you would need to explain what these wards are (methods section).

Have added literature and amended

6. General comments

Throughout the manuscript there is a lack of correct use of capitals and mixed use of terminology that

confuses the reader eg. use of Steering Group and Stakeholder Reference Group- are these the same? Consultant and Old Age Psychiatrist. This needs to be corrected throughout. Tables need to be tidied to ensure consistent headings and that where * have been used - a descriptor is provided. Remove or amend empty sections of table.

Thanks, comment noted and amended accordingly

There was only one group, we refer to it as the Steering Group throughout

I wish you all the best with your revision

Reviewer: 2

Dr. Karen Mate, University of Newcastle

Thanks, noted and amended text and appendix table 5

Source of each variable is presented in appendices 1 and 2

- There is no specific explanation of how missing data were addressed. The predictor variables related to patients receiving care according to audit standards in particular, have high numbers of missing data. For example, "discharge planning within 24 hours of admission" has data missing for 26.9% of patients and N/A for 20.7% of patients. The authors should provide an explanation of the "N/A" category, whether these data were included in the analyses or not, and how missing data was managed in general.

Thanks. Noted, added some details to the text and below from study statistician:

The main analysis was made on the complete data so missing data wasn't given any treatment

You could do multiple imputation (it is very fashionable nowadays) and then sensitivity analysis. However, this is very complex, especially in this case where there are explanatory variables at the hospital and patient level. And it's valid under certain assumption which are difficult to verify

Results

- Table 1: for interaction term (old age liaison psychiatrist), specify categories "yes vs no" or "present vs absent"
- Table 2: title should specify that the summary provided is for "organisation/delivery" or "modifiable?" predictors only; patient level and ward level factors are not included

Amended

Discussion

- The statement "Of these variables, planning discharge within 24 hours of admission had the strongest association with length of hospital stay." is not true. There are other non-modifiable or patient factors that have similar or stronger negative association with length of hospital stay (ie associated with shorter stay), and evidence of discussing discharge has similar positive association (ie associated with shorter stay). The statement is making a key point but needs to be modified so that it is correct eg "Of the modifiable (or organisational/delivery related) variables, planning discharge had the strongest negative association with length of hospital stay".
- Make the direction of associations more clear in the Discussion.
- Some of the Discussion of published literature (the authors own studies) is written as if it is part of the current study eg "in our interviews....." and "staff told us....." which is not appropriate. This is particularly the case when dealing with the association between carer involvement and longer admissions. The authors should make a clear statement about the findings of the present study and then proceed to elaborate on the published literature (including their own) to provide possible explanations of the findings.

Thanks, amended

- It would be helpful for the authors to comment on the magnitude of the associations and whether they are of clinical significance in terms of the length of hospital stay.

I suspect that what you would like to see is standardized coefficients. However, this only makes sense when there are no interactions. What one can do is to interpret the results of the effects of the explanatory variables in terms of their units. Obviously, this doesn't apply when they are categorical.

Example: a one unit increase in age; the average length of stay decreases by 0.3 percent.

Finally, there are few typos throughout the text eg

Pg 4 "symotms"

Pg10 "mor"

Pg11 "reqite"

Thanks, amended

VERSION 2 – REVIEW

REVIEWER	Mate, Karen University of Newcastle, School of Biomedical Sciences and Pharmacy
REVIEW RETURNED	15-Jun-2021
GENERAL COMMENTS	The authors have addressed the issues raised by reviewers